# Investigating the Impact of Data Distribution Shifts on Crossmodal Knowledge Distillation

## Abstract

Cross-modal knowledge distillation (KD) has expanded the traditional KD approach to encompass multimodal learning, achieving notable success in various applications. However, in cases where there is a considerable shift in data distribution during cross-modal KD, even a more accurate teacher model may not effectively instruct the student model. In this paper, we conduct a comprehensive analysis and evaluation of the effectiveness of cross-modal KD, focusing on its dependence on the distribution shifts in multimodal data. We initially view cross-modal KD as training a maximum entropy model using pseudo-labels and establish conditions under which it outperforms unimodal KD. Subsequently, we introduced the hypothesis of solution space divergence, which unveils the crucial factor influencing the efficacy of cross-modal KD. Our key observation is that the accuracy of the teacher model is not the primary determinant of the student model's accuracy; instead, the data distribution shifts play a more significant role. We demonstrate that as the data distribution shifts decrease, the effectiveness of cross-modal KD improves, and vice versa. Finally, to address significant data distribution differences, we propose a method called the "perceptual solution space mask" to enhance the effectiveness of cross-modal KD. Through experimental results on four multimodal datasets, we validate our assumptions and provide directions for future enhancements in cross-modal knowledge transfer. Notably, our enhanced KD method demonstrated an approximate 2% improvement in *mIoU* compared to the Baseline on the SemanticKITTI dataset.

## 1 Introduction

Knowledge distillation (KD) is an effective technique for transferring knowledge from one neural network to another. The core idea behind KD is to align the predicted logits of a teacher network with those of a student network by minimizing the Kullback-Leibler divergence (KL Divergence (Hinton et al., 2015)). This concept can be easily extended to handle multimodal data by using a multi-branch network structure, where the teacher and student networks can take input from single or multimodal data sources (Gou et al., 2021).

KD is commonly applied in three scenarios. Firstly, in the context of knowledge expansion or knowledge transfer, cross-modal KD is used to overcome limitations in a single modality. For example, it can compensate for deficiencies such as insufficient data volume, lack of depth information in RGB images, or absence of texture information in point cloud data. Secondly, KD enables multimodal knowledge fusion by transferring knowledge from one modality to another. This allows leveraging the knowledge of a multimodal teacher network while using a single modality during testing. Finally, KD can provide additional constraints in multimodal scenarios, including cross-modal retrieval and cross-domain adaptation.

While cross-modal KD has achieved success in various multimodal data applications, most widely used methods still rely primarily on single-modal approaches. This raises questions about the effectiveness of cross-modal KD and the factors that limit its performance. Recent research focusing on the effectiveness of cross-modal KD indicates that its performance depends on the modality-general decisive features retained in the teacher network (Xue et al., 2023). These features characterize the alignment level between different modalities, and a higher degree of feature alignment leads to better KD results. Fig.1(a) shows an example of a multimodal dataset that includes both sound and

Figure 1: Some Cross-Modal Data Instances.(a) Modal misaligned scene (b)-(d) Modal alignment scenario (Our research subjects), such as images and audio of a guitar, RGB images from the same camera perspective and point cloud projected onto images and depth maps.

images. In this scenario, the image data might not only capture the guitar playing music but also a considerable amount of background information. As a result, the information between the two modalities is not entirely aligned. According to the modality focus hypothesis (Xue et al., 2023), the effectiveness of cross-modal distillation is expected to increase with better alignment of features between the two modalities. However, it has been discovered that even in cases of perfect alignment between modalities, cross-modal KD can still be ineffective, as illustrated in Fig.1(b), Fig.1(c) and Fig.1(d).

In contrast to (Xue et al., 2023), this paper focuses on examining the impact of data distribution shifts on cross-modal KD. We introduce the Solution Space Divergence Hypothesis (SSDH) to explain the challenges faced by cross-modal KD. Furthermore, to address instances of failed cross-modal KD, we propose a method known as the Perceptual Solution Space Mask (PSSM), which enhances the effectiveness of cross-modal KD.

The main contributions of our paper can be summarized as follows:

- We establish the conditions under which cross-modal KD outperforms unimodal scenarios.
- We propose the SSDH, which suggests that the effectiveness of cross-modal KD is determined by the differences in the distribution of input modalities.
- To substantiate the validity of the SSHD, we propose a method called PSSM. This approach enhances the performance of cross-modal KD by masking positions where the solution spaces of the teacher and student networks are inconsistent, thereby validating our proposed hypothesis.
- We conducted experiments on four cross-modal datasets, and the results validate our hypothesis and the effectiveness of the proposed method.

## 2 RELATED WORK

### 2.1 CROSSMODAL KD

With the widespread use of the internet and the increasing application of multimodal sensors, there has been a surge of interest in multimodal learning. In line with this trend, KD has evolved from its original purpose of model compression to encompass knowledge transfer in multimodal data settings, such as semantic segmentation (Li et al., 2020; Chen et al., 2019; Li et al., 2020; Yan et al., 2022; Jaritz et al., 2022; Li et al., 2020; Yan et al., 2022), 3D human pose estimation (Thoker & Gall, 2019), emotion recognition (Li et al., 2023), super-resolution reconstruction (Xia et al., 2022), and object detection (Dai et al., 2021; Zhao et al., 2020; Chen et al., 2022; Abavisani et al., 2019; Yang & Xu, 2021; Zhang & Ma, 2020; Yao et al., 2022), among others (Huang et al., 2022). The mentioned methods commonly utilize paired multimodal data,

where one modality or a combination of modalities is employed to train the teacher network. Simultaneously, the data from another modality is used as the training data for the student network. Through the minimization of the divergence between the output distributions of the teacher and student models, the student model can effectively acquire knowledge from the teacher model. This process enhances the robustness and overall performance of the student model.

Additionally, certain methods have explored the use of non-paired data for training purposes (Dou et al., 2020; Wang et al., 2021). Although these approaches have shown promise in cross-modal KD, they tend to be tailored to specific multimodal tasks. This paper, on the other hand, primarily focuses on conducting a thorough analysis of the underlying mechanisms in cross-modal KD. By delving deeper into these mechanisms, the aim is to gain a better understanding of cross-modal KD and its potential implications.

## 2.2 KD ANALYSIS

while knowledge distillation (KD) has demonstrated success across various domains, there are instances where its effectiveness may not be satisfactory, leading to distillation failures. Consequently, researchers have made efforts to explore and study the factors influencing KD. Among these factors, the structure and capacity of both the teacher and student networks have received considerable attention from researchers (Cho & Hariharan, 2019). For example (Mirzadeh et al., 2020) employed a medium-sized network, often referred to as a 'teacher assistant,' to bridge the gap between the student and the teacher. (Li et al., 2017) proposed a novel algorithm based on the distillation process for learning from noisy data. Additionally, (Ren et al., 2022) and (Mirzadeh et al., 2020) explored the transfer of inductive biases of different networks, such as Long Short-Term Memory(Hochreiter & Schmidhuber, 1997), Convolutional Neural Network(CNN)(LeCun et al., 1989), or Multi-Layer Perceptron(MLP), through KD. Some other researchers have analyzed the poor performance of KD for specific tasks and proposed improvement strategies. For instance, (Guo et al., 2021) highlighted the importance of including background information without objects in object detection tasks for KD. (Yang et al., 2020) advocated KD based on representation learning, which involves minimizing the difference between the penultimate layer feature representations of the teacher and student networks. In (Abavisani et al., 2019), KD was combined with model quantization to compress models, resulting in enhanced performance. Recently, (Xue et al., 2023) investigated the impact of the concentration of cross-modal features on KD. In contrast to (Xue et al., 2023), our focus in this paper is on the influence of data distribution shift between cross-modal data on KD.

## 2.3 DATA DISTRIBUTION SHIFTS

Data distribution shifts has become a critical factor influencing the fusion of multimodal data. To tackle the challenge of domain shift, two broad approaches have been pursued. One approach focuses on decoupling and aligning features to achieve multimodal data fusion (Li et al., 2023; Zhao et al., 2020). These methods separate multimodal features into modality-related and modality-unrelated components and subsequently align the related features. The other approach involves unsupervised adversarial learning, where adversarial techniques are employed to transform features from one modality to another or to an intermediate modality (Li et al., 2020). However, these methods primarily address the fusion of multimodal features and may not be directly applicable to dual-branch network structures in the context of cross-modal KD. To the best of our knowledge, there is currently no existing literature that specifically addresses the adverse effects of domain shift on cross-modal KD. In this paper, we propose a mask-based approach to alleviate the failure of cross-modal KD caused by data distribution shift issues.

## 3 THE PROPOSED APPROACH

### 3.1 CONDITIONAL ASSUMPTIONS AND SYMBOL DEFINITIONS

In this paper, we begin by providing a comprehensive overview of the fundamentals of KD and introducing the symbol representations utilized throughout the study. Our focus is primarily on the case of C-class classification, although the concepts discussed can be extended to regression tasks as well. To maintain a general framework, we consider a two-modal setting, where the data is represented as $x^a$ and $x^b$, denoting the data from modalities 'A' and 'B', respectively.

$$\begin{cases} \{(x_i^a, y_i)\}_{i=1}^n \sim P^n(x^a, y), x_i^a \in \mathbb{R}^{d_a}, y_i \in \Delta^c \\ \{(x_i^b, y_i)\}_{i=1}^n \sim P^n(x^b, y), x_i^b \in \mathbb{R}^{d_b}, y_i \in \Delta^c \end{cases} \tag{1}$$

Here, $(x_i^a, y_i)$ and $(x_i^b, y_i)$ respectively represent the feature-label pair of modality 'A' and modality 'B'. and $\Delta^c$ represents the set of c-dimensional probability vectors. Suppose our goal is to train a student network that takes $x^b$ as input. In the case of cross-modal KD, the teacher network takes the training data $x^a$ as its input and minimizes the training objective:

$$f_t = \underset{f \in \mathcal{F}_t}{\arg\min} \frac{1}{n} \sum_{i=n}^{n} \mathcal{L}(y_i, \sigma(f(x_i^a))) + \Omega(\|f\|) \tag{2}$$

Where $\mathcal{F}_t$ is a class of functions from $\mathbb{R}^{d_a}$ to $\mathbb{R}^c$, the function $\sigma : \mathbb{R}^c \to \Delta^c$ is the softmax operation.

$$\sigma(z)_k = \frac{e^{zk}}{\sum_{j=1}^{c} e^{z_j}} \tag{3}$$

For all $1 \le k \le c$, the function $\mathcal{L} : \Delta^c \times \Delta^c \to \mathbb{R}$ is KL Divergence. And $\Omega : \mathbb{R} \to \mathbb{R}$ is an increasing function which serves as a regularizer. *Note that here the KL divergence is equivalent to the cross-entropy(CE) loss.*

After training the teacher model $f_t$ using the data $x_i^a$ from modality 'A', our goal is to transfer the knowledge acquired by the teacher network to the student network operating in modality 'B'. The objective in optimizing the student network is to minimize Eqn. (4).

$$f_s = \underset{f \in \mathcal{F}_s}{\arg\min} \frac{1}{n} \sum_{i=1}^{n} [(1-\lambda) \cdot \underbrace{\mathcal{L}(y_i, \sigma(f(x_i^b)))}_{CE} + \lambda \cdot \underbrace{\mathcal{L}(s_i, \sigma(f(x_i^b)))}_{KL}] \tag{4}$$

Where $s_i = \sigma(f_t(x_i^a)/T) \in \Delta^c$ represents the soft predictions obtained from $f_t$ about the training on modality 'A'. The temperature parameter $T$ $(T > 0)$ controls the level of softening or smoothing of the class-probability predictions from $f_t$ and the imitation parameter $\lambda \in [0, 1]$ determines the balance between imitating the soft predictions $s_i$ and predicting the true hard labels $y_i$. *Note that as $s_i$ represents predicted labels rather than true labels, the loss function $\mathcal{L}$ used at this stage cannot be equivalent to the CE loss typically used for training with true labels.*

Given that our primary focus in this paper is the examination of data distribution shift effects concerning cross-modal KD, we delineate the ensuing assumption conditions in our discourse. These assumptions form the foundational framework for the theoretical elucidations in Sec. 3.2.

- **Assumption 1:** $\mathcal{F}_s$ and $\mathcal{F}_t$ have the same model capacity, meaning they have the same ability to fit or learn complexity or accommodate information.

- **Assumption 2:** $x_i^a$ and $x_i^b$ have the same modality strength, meaning that when the same model is trained using $x_i^a$ and $x_i^b$ as data separately, the difference in model prediction accuracy is not significant.

- **Assumption 3:** Considering the vector output from Eqn. (3), which encompasses the model's scores or probability estimates for each potential class, and assuming that assumptions 1 and 2 are satisfied, the key distinction between cross-modal knowledge distillation and single-modal knowledge distillation emerges from the variability in data distribution. Consequently, we have the option to substitute the solution space, initially characterized by network parameters, with the probability distribution output after feeding the data into the network.

## 3.2 SOLUTION SPACE DIVERGENCE HYPOTHESIS

Under the assumption in 3.1, based on VC theory (Vapnik, 1999), we can prove that: **Unimodal KD serves as an upper bound for cross-modal KD.** (See Appendix A for the omitting proof.) According to the above conclusion, we can infer that the shift in data distribution is a crucial factor affecting the effectiveness of cross-modal KD. This data distribution shift ultimately leads to the inconsistency of solution spaces between the teacher and student networks, resulting in the teacher's inability to guide the student. This gives rise to the Solution Space Divergence Hypothesis (SSDH).

**Solution Space Divergence Hypothesis (SSDH)**.

*For cross-modal KD, the performance of KD is determined by the Divergence in the solution space between the teacher and student networks, which is caused by the input modality. The smaller the Divergence in solution space, the better the student network is expected to perform.*

This hypothesis posits that in cross-modal knowledge transfer, the student learns to 'focus on' features in the teacher network that align with its own network's solution space. Therefore, cross-modal KD benefits from the consistency of solution spaces. Furthermore, it elucidates our observation that, in certain circumstances, the teacher's performance is not directly correlated with the student's performance.

To intuitively and quickly comprehend our hypothesis, we conducted an experiment using synthetic Gaussian data in this context. We maintained a constant input for the student network and generated 6 sets of cross-modal data by varying the input data for the teacher network, thereby satisfying **Assumption 2**. Both the teacher and student networks employed the same 2-layer MLP, in accordance with **Assumption 1**. To better characterize the differences in solution spaces between different data modalities, as stipulated by **Assumption 3**, we quantified the discrepancy between the solution spaces of the student and teacher networks using Central Moment Discrepancy (CMD) (Zellinger et al., 2017), based on the Moment Matching Theorem. A higher CMD value indicates a greater disparity in solution spaces.

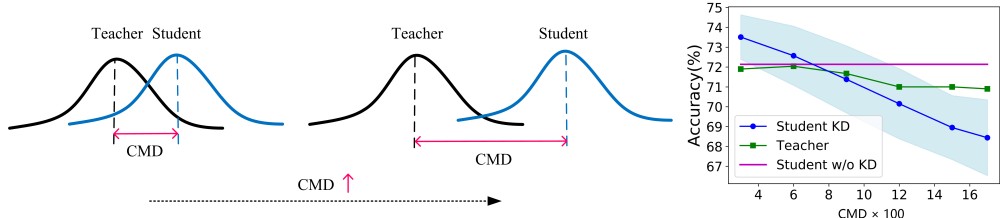

Figure 2: The teacher modality is $x_i^a$, and the student modality is $x_i^b$. We plot a confidence interval of one standard deviation for student accuracy. As CMD increases, cross-modal KD becomes ineffective.

As shown in Fig. (2), we begin by considering scenarios where the disparity in solution spaces between the two modalities was at its maximum. By gradually increasing shared deterministic features, we progressively reduced the dissimilarity in solution spaces. We observed that when there was a significant disparity in solution spaces between the teacher and student, cross-modal KD did not function effectively, as the approximation error was substantial at this point. As the dissimilarity in solution spaces decreased, cross-modal KD became more effective. It's important to note that throughout this process, the teacher's accuracy remained relatively stable, while there was a substantial variation in student performance.

## 3.3 PERCEPTUAL SOLUTION SPACE MASK (PSSM)

In accordance with the hypothesis in Sec. 3.2, we have demonstrated that in cross-modal KD, the inconsistency in solution spaces leads to differences between the student and teacher prediction distributions, ultimately resulting in an increase in approximation error. To mitigate this decline in KD due to multimodal data, in this section, we propose a mask-based approach called the Perceptual Solution Space Mask(PSSM) to enhance cross-modal KD.

Building upon the analysis above, we introduce the concept of a perceptual solution space inconsistency mask to focus on output features that arise from differences in solution spaces. Our approach involves computing a mask for each pair of teacher and student network-predicted output features, where features with larger differences in solution spaces are assigned smaller weights, and vice versa. The mask values range from 0 to 1. As features with greater dissimilarity in solution spaces receive smaller weights, this approach can mitigate the impact of solution space inconsistency. Specifically, we first calculate the difference between the predicted distributions of the student and teacher in the final classification output layer. Then, based on the computed distribution discrepancy, we obtain the specific mask values.

To compute the difference between probability distributions of different modalities' outputs, we first calculate the cosine similarity between the distributions.

$$distance = \frac{1}{2} \cdot \left( \frac{\mathbf{O}^a \cdot \mathbf{O}^b}{\|\mathbf{O}^a\| \|\mathbf{O}^b\|} + 1 \right) \tag{5}$$

Here, $\|\cdot\|$ represents the L2 norm. To normalize the distribution discrepancy in our computed results to a range of 0 to 1, we add 1 to the calculated cosine similarity and then divide it by 0.5. Furthermore, lower similarity in output distributions indicates greater dissimilarity between modalities. By combining this with a cosine distance threshold, we obtain the mask:

$$mask = \begin{cases} 1, & if \quad distance < \tau \\ distance, & otherwise \end{cases} \tag{6}$$

Here $\tau$ indicates the distance threshold. Finally, for the last output layer, the maximum value in the predicted probability distribution should align with the one-hot label; otherwise, the prediction distribution is definitely incorrect. Based on this prior information, we only apply the mask when the predicted label is incorrect.

$$\Omega_i^{a \rightarrow b} = \begin{cases} 1, & if \quad predict = label \\ mask, & otherwise \end{cases} \tag{7}$$

Inspired by (Zhang et al., 2018), to learn perceptual information from the data distributions of different modalities, we construct the perception-aware loss with respect to cross-modal KD loss by

$$\mathcal{L}_{per} = \Omega_i^{a \rightarrow b} \cdot KL_{multi}(\mathcal{F}_t||\mathcal{F}_s) \tag{8}$$

Because this method operates based on the input data at the final layer and does not involve modifications to the network model itself, it can be easily ported to other improved KD algorithms. The details will be discussed in Sec. 4.3.

# 4 EXPERIMENTAL RESULTS

## 4.1 EXPERIMENTAL SETUP

To validate our SSDH and evaluate PSSM, we conducted four experiments on multimodal datasets. We employ three methods to construct multimodal data with domain shifts. (1) For synthetic Gaussian and Sklearn datasets, we kept the input features fixed for the student network and generated multimodal data with varying degrees of modality differences by replacing teacher features with different numbers of student features. Detailed information can be found in the Appendix. A. It's important to note that, on the synthetic Gaussian dataset, we primarily aimed to validate the soundness of SSDH and PSSM. On the Sklearn Dataset, we applied the PSSM to the DKD loss (Zhao et al., 2022), significantly enhancing its performance in cross-modal KD. This demonstrates the strong generalization capability of our method. For more details, please refer to Sec. 4.3. (2) For the MNIST-MNISTM dataset, we employed MNISTM and MNIST data as the inputs for the student and teacher networks, respectively. To induce distribution shifts, we introduced varying levels of noise to the MNIST data. (3) For the SemanticKITTI dataset, we used point clouds and RGB images as inputs for the student and teacher networks, respectively. We modified the input data distribution of the teacher network by incorporating point cloud features into the image features.

## 4.2 SYNTHETIC GAUSSIAN

In line with the settings provided by the (Xue et al., 2023), we expand upon the Gaussian example presented in (Lopez-Paz et al., 2015) to encompass a multimodal scenario (see Appendix **??** for details). This dataset represents a binary classification task, consisting of six pairs of data distributions with distinct differences. This type of data distribution shift leads to an inconsistency in

solution spaces, as illustrated in Fig. (2).These findings align with the perspectives of the SSDH, which posits that minimizing the disparity between teacher and student solution spaces fosters improved student performance. To further analyze the origin of this inconsistent solution space, we computed the CMD distances between the teacher and student networks regarding their input data and the CMD distances between their output probability distributions, as shown in the first and second columns of Table. 1, respectively. Observably, as the CMD value of the input data increases, the CMD value of the network outputs also increases, signifying that the deviation in data distribution results in an inconsistency in the solution space. It is worth noting that in the experiments, we solely manipulated the input data while keeping other variables constant. When we further visualize the probability distributions of the teacher and student network outputs, we notice that an increase in the disparity of input data distribution (indicated by the increase in CMD) corresponds to an enlargement of the solution space differences, as depicted in Fig. (3).

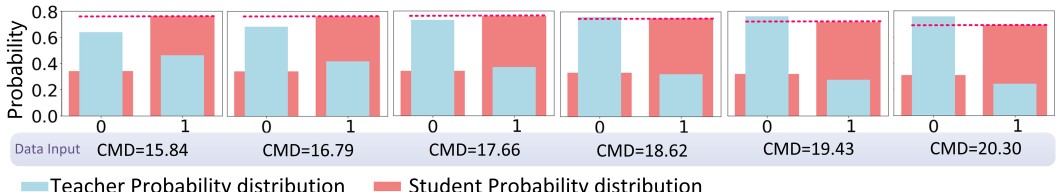

Figure 3: Visualizing the Probability Distributions of Teacher and Student Network Outputs When Teacher Predicts Incorrectly.

To address the problem of KD failure caused by significant disparities in the solution spaces of teachers and students, we employed the PSSM method proposed in Sec. 3.3. A comparison of the results from Table. 1 and Fig. (4) 1 reveals that the method incorporating the perceptual solution space mask achieves lower CMD values compared to the original method. This observation suggests that our approach effectively mitigates the performance degradation caused by solution space disparities. Upon further examination, we have noticed that under conditions where solution space disparities are minimal, the improvements achieved by the enhanced algorithm are limited and, in some cases, do not surpass the performance of the original KD method (e.g., CMD=3). This further emphasizes that the performance improvement in PSSM is not due to the elimination of inaccurate pseudo-labels but rather the reduction in solution space divergence. *The code can be found in the supplementary materials.*

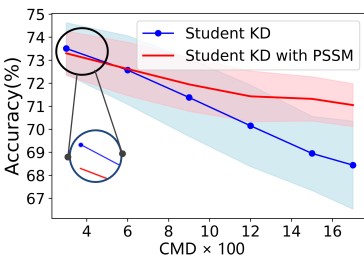

Figure 4: Using KL as the baseline, compare the accuracy of the student network predictions with and without PSSM.

Table 1: The CMD distance between the teacher and student networks for input data, as well as the CMD distance between their output probability distributions.

| Input Data | w/o PSSM | w/ PSSM |
|---|---|---|
| 15.84 | 0.03 | 0.03 - |
| 16.79 | 0.06 | 0.05 ↓ |
| 17.66 | 0.09 | 0.07 ↓ |
| 18.62 | 0.11 | 0.08 ↓ |
| 19.43 | 0.13 | 0.09 ↓ |
| 20.30 | 0.15 | 0.11 ↓ |

### 4.3 SKLEARN DATASETS

In this section, we extend the PSSM to (Zhao et al., 2022), this approach reformulates the classical KD loss into two distinct components: Target Class Knowledge Distillation (TCKD) and Non-Target Class Knowledge Distillation (NCKD). TCKD represents the similarity of binary probabilities for target classes between the teacher and student networks. Meanwhile, NCKD represents the similarity of probabilities between the teacher and student networks for non-target classes. The DKD loss with the PSSM is formulated as follows:

$$\text{DKD}_{PSSM} = \Omega \cdot \text{DKD} \tag{9}$$

Where $\Omega$ is determined by Eqn. (7), $\text{DKD} = \alpha\,\text{TCKD} + \beta\,\text{NCKD}$ and $\alpha$, $\beta$ are hyperparameters used to balance the importance of TCKD and NCKD. For details, please refer to (Zhao et al. (2022)).

Table. 2 presents a comparison between the results of DKD with PSSM and the original DKD loss. From the experimental results, we observe a significant performance improvement of DKD compared to the KD loss, with an enhancement of approximately 3%, when the solution space difference is relatively small (i.e., a smaller CMD value). However, as the solution space shifts increases, DKD performance deteriorates notably, even exhibiting negative distillation effects. Interestingly, we also discover that when we extend PSSM to DKD, there is a notable improvement in accuracy, indicating the effectiveness of our method in enhancing cross-modal KD.

In addition, we investigated the sensitivity of TCKD and NCKD to inconsistent solution spaces by varying $\alpha$ and $\beta$. From Table. 3, we observe that when $\beta$ is fixed, increasing $\alpha$ leads to effective KD regardless of whether the solution space difference increases. However, when $\alpha$ is fixed and $\beta$ is increased, the KD performance significantly deteriorates, especially when the solution space difference is large (indicated by a larger CMD value), as shown in Table. (4). This suggests that in the context of cross-modal KD, NCKD is more sensitive to the inconsistency in solution spaces compared to TCKD.

Table 2: Results on Sklearn datasets. $\alpha = 100, \beta = 1$

| CMD Distance | Teacher Accuracy | Student w/o KD | Student KD KL | Student KD DKD | Student KD DKD with PSSM |
|---|---|---|---|---|---|
| 4.20 | 41.36 | 41.73 | 42.61 $(0.88 \pm 0.40)$ | 46.46 $(4.73 \pm 1.39)$ | 48.54 $(6.81 \pm 1.18)$ |
| 4.68 | 42.71 | 41.73 | 42.28 $(0.55 \pm 0.47)$ | 45.87 $(4.14 \pm 2.87)$ | 48.61 $(6.89 \pm 2.26)$ |
| 5.15 | 39.88 | 41.73 | 41.83 $(0.10 \pm 0.73)$ | 42.07 $(0.34 \pm 3.17)$ | 46.39 $(4.67 \pm 2.39)$ |
| 5.60 | 40.46 | 41.73 | 41.68 $(-0.05 \pm 0.69)$ | 41.12 $(-0.60 \pm 3.37)$ | 44.93 $(3.20 \pm 2.84)$ |
| 5.96 | 42.36 | 41.73 | 41.53 $(-0.19 \pm 0.42)$ | 40.98 $(-0.74 \pm 3.06)$ | 44.82 $(3.09 \pm 2.70)$ |

Table 3: Results on Sklearn datasets. Fixing $\beta = 1$ and varying $\alpha$. The data in the table shows the difference in prediction accuracy when using DKD with PSSM compared to not using KD. A larger value indicates a better improvement.

| CMD \ $\alpha$ | 1 | 4 | 8 | 16 | 20 |
|---|---|---|---|---|---|
| 5.96 | $0.06 \pm 0.45$ | $0.06 \pm 1.06$ | $0.01 \pm 1.42$ | $0.65 \pm 1.97$ | $0.92 \pm 2.16$ |
| 5.60 | $0.42 \pm 0.49$ | $0.45 \pm 0.77$ | $0.57 \pm 1.08$ | $1.16 \pm 1.64$ | $1.52 \pm 1.74$ |
| 5.51 | $0.29 \pm 0.39$ | $0.94 \pm 1.06$ | $1.53 \pm 1.48$ | $2.57 \pm 1.80$ | $2.95 \pm 1.87$ |
| 4.68 | $0.50 \pm 0.43$ | $1.79 \pm 1.03$ | $2.89 \pm 1.44$ | $4.11 \pm 1.50$ | $4.61 \pm 1.44$ |
| 4.20 | $0.77 \pm 0.31$ | $1.85 \pm 0.61$ | $2.92 \pm 1.13$ | $3.91 \pm 1.20$ | $4.27 \pm 1.26$ |

Further analysis of this phenomenon reveals that, due to the common constraint of both teacher and student networks on target classes using the same CE loss, there is a significant bias in probability prediction for non-target classes due to the shift in data distribution (resulting in larger solution space divergence). Therefore, the increase in TCKD, which represents the similarity of probabilities for target classes between the teacher and student, helps in suppressing NCKD, which represents the similarity of probabilities for non-target classes. This demonstrates that enhancing TCKD, which reflects the similarity of target class probabilities, contributes to the suppression of NCKD, which represents the similarity of non-target class probabilities.

Table 4: Results on Sklearn datasets. Fixing $\alpha = 1$ and varying $\beta$. The data in the table shows the difference in prediction accuracy when using DKD with PSSM compared to not using KD. A larger value indicates a better improvement.

| CMD \ $\beta$ | 1 | 4 | 8 | 16 | 20 |
|---|---|---|---|---|---|
| 5.96 | 0.06 ± 0.45 | -0.58 ± 0.73 | -1.79 ± 1.28 | -3.94 ± 2.22 | -4.98 ± 2.60 |
| 5.60 | 0.42 ± 0.49 | 0.04 ± 0.84 | -0.81 ± 0.82 | -2.03 ± 1.23 | -2.60 ± 1.49 |
| 5.15 | 0.29 ± 0.39 | 0.25 ± 0.69 | -0.18 ± 0.94 | -0.95 ± 1.07 | -1.45 ± 1.28 |
| 4.68 | 0.50 ± 0.43 | 0.75 ± 0.73 | 0.70 ± 0.97 | 0.83 ± 1.20 | 0.73 ± 1.27 |
| 4.20 | 0.77 ± 0.31 | 1.10 ± 0.53 | 1.43 ± 0.76 | 2.09 ± 0.92 | 2.34 ± 1.01 |

## 4.4 MNIST/MNISTM AND SEMANTICKITTI DATASETS

We applied the proposed PSSM to both simulated and real-world datasets to further validate the effectiveness of PSSM.

**Experimental Results on MNIST/MNISTM Dataset**: Table. (5) presents the comparative experimental results of using DKD loss and our DKD with PSSM under different noise levels. From Table. 5, we can observe a significant improvement in accuracy when applying the PSSM. Particularly, under high noise levels, our method demonstrates substantial performance enhancement. For instance, at a noise level of 5, our approach achieved an approximately 6% increase in classification accuracy.

Table 5: Results on MNIST/MNISTM dataset. $\alpha = 16, \beta = 1$

| Noise level | Teacher Accuracy | Student w/o KD | Student KD DKD | Student KD DKD with PSSM | $\Delta$ |
|---|---|---|---|---|---|
| 0 | 92.98 | 75.40 | **82.03**(6.62 ± 0.67) | 81.83 (6.64 ± 0.76 ) | -0.2 |
| 1 | 81.10 | 75.40 | 79.87 (4.74 ± 0.72) | **81.21** (5.81 ± 0.88) | +1.34 |
| 2 | 70.42 | 75.40 | 76.52 (1.12 ± 2.17) | **78.77** (3.37 ± 0.81) | +2.25 |
| 3 | 62.58 | 75.40 | 75.14 (-0.26 ± 2.44) | **79.92** (4.52 ± 1.28) | +4.78 |
| 4 | 46.37 | 75.40 | 72.19 (-3.21 ± 1.87) | **76.94** (1.52 ± 0.78) | +4.75 |
| 5 | 38.85 | 75.40 | 68.99 (-6.17 ± 5.66) | **76.20** (0.66 ± 1.15) | +7.21 |

**Experimental Results on SemanticKITTI Dataset**: We compared the point cloud segmentation results using KD loss, DKD, and the improved DKD loss, as shown in Table. (6). It can be observed that without incorporating point cloud information, both KD loss and DKD loss are ineffective. However, our method still achieves a performance improvement of 1.5%, indicating its effectiveness in overcoming the negative impact caused by data distribution shift.

Table 6: Results on SemanticKITTI.

| | Teacher mIoU(%) | Student w/o KD mIoU (%) | Student KD mIoU(%) | Student KD+PSSM mIoU(% |
|---|---|---|---|---|
| a | 46.04 | 59.68 | 60.19 | 62.40 |
| b | 61.7 | 59.68 | 62.95 | 63.90 |

## 5 CONCLUSION AND LIMITATIONS

In this study, we conducted a comprehensive exploration of cross-modal KD and its broader applications in multimodal learning. We introduce the SPDH, highlighting the role of data distribution disparities across modalities in KD effectiveness. Additionally, we propose PSSM to mitigate the impact of data distribution shifts on cross-modal KD. However, it's important to note that our investigation primarily centered on cross-modal KD rooted in logit distillation, omitting the exploration of alternative approaches based on distilling deep features from intermediate layers. Future research endeavors may expand upon these aspects to further advance cross-modal KD in the academic community.

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

## A   PROOF OF THEOREM

| Symbol List | |
|---|---|
| $x^a$ | data from modalities 'A' |
| $x^b$ | data from modalities 'B' |
| $\mathcal{F}_t$ | teacher function (trained on $x^a$) |
| $\mathcal{F}_s$ | student function (trained on $x^a$ or $x^b$) |
| $T$ | temperature parameter |
| $y_i$ | the true hard labels |
| $s_i$ | soft predictions |
| $\varepsilon$ | approximation error |
| $O(\cdot)$ | Estimation error |
| $\sigma$ | the softmax operation. |
| $|\cdot|_C$ | some function class capacity measure |
| $n$ | The number of data point |

Recall our three actors: the student function $f_s \in \mathcal{F}_s$ (trained on $x_i^b$), the teacher function $f_t \in \mathcal{F}_t$ (trained on $x_i^a$ or $x_i^b$), and the real target function of interest to both the student and the teacher, $f \in \mathcal{F}$. For simplicity, consider pure distillation, where the imitation parameter is set to $\lambda = 1$.

According to VC theory (Vapnik, 1999), the classification error of the classifier, $f_s^b$ can be expressed as:

$$R(f_s^b) - R(f) \leq O\left(\frac{|\mathcal{F}_s^b|_C}{\sqrt{n}}\right) + \varepsilon_{sb} \tag{10}$$

Where the $O(\cdot)$ and $\varepsilon_{sb}$ terms are the estimation and approximation error, respectively. The former refers to the performance gap between a model on training data and its theoretical best performance. The latter refers to the difference between a model output and the true target function. It reflects whether the model representational capacity is sufficiently powerful to accurately approximate the true target function. If the model's hypothesis space is not capable of capturing the complexity of the target function, the approximation error will be large. Here $R$ is the error, $|\cdot|_C$ is some function class capacity measure, and $n$ is the number of data point.

Let $f_t^a \in \mathcal{F}_t^a$ and $f_t^b \in \mathcal{F}_t^b$ be the teacher function trained on $x_i^a$ and $x_i^a$, then:

$$R(f_t^a) - R(f) \leq O\left(\frac{|\mathcal{F}_t^a|_C}{n}\right) + \varepsilon_{ta} \tag{11}$$

$$R(f_t^b) - R(f) \leq O\left(\frac{|\mathcal{F}_t^b|_C}{n}\right) + \varepsilon_{tb} \tag{12}$$

Then, we can transfer the knowledge of the teacher separately from training data 'A' or 'B' to the student. Let $f_t^a$ serve as the teacher function in cross-modal KD, and $f_t^b$ in Unimodal KD, then:

***Cross modal KD***

$$R(f_s^b) - R(f_t^a) \leq O\left(\frac{|\mathcal{F}_s^b|_c}{n^\alpha}\right) + \varepsilon_m \tag{13}$$

Where $\varepsilon_m$ is the approximation error of the teacher function class $\mathcal{F}_s^b$ with respect to $f_t^a \in \mathcal{F}_t^a$, and $\frac{1}{2} \leq \alpha \leq 1$.

***Unimodal KD***

$$R(f_s^b) - R(f_t^b) \leq O\left(\frac{|\mathcal{F}_s^b|_C}{n^\alpha}\right) + \varepsilon_l \tag{14}$$

Where $\varepsilon_l$ is the approximation error of the teacher function class $\mathcal{F}_s^b$ with respect to $f_t^b \in \mathcal{F}_t^b$.

Then, if we employ cross-modal KD, combining Eqn. (11) and Eqn. (13), we can obtain an alternative expression for the student learning the real function $f$, as follows:

***Alternative Expression Through Cross-Modal KD***

$$
\begin{aligned}
R(f_s^b) - R(f) &= R(f_s^b) - R(f_t^a) + R(f_t^a) - R(f) \\
&\leq O\left(\frac{\left|\mathcal{F}_s^b\right|_C}{n^\alpha}\right) + \varepsilon_m + O\left(\frac{\left|\mathcal{F}_t^a\right|_C}{n}\right) + \varepsilon_{ta}
\end{aligned}
\tag{15}
$$

Similarly, by employing unimodal KD and combining Eqn. (12) and Eqn. (14), we can obtain an alternative expression for the student's learning of the real function $f$, as follows:

***Alternative Expression Through Unimodal KD***

$$
\begin{aligned}
R(f_s^b) - R(f) &= R(f_s^b) - R(f_t^b) + R(f_t^b) - R(f) \\
&\leq O\left(\frac{\left|\mathcal{F}_s^b\right|_C}{n^\alpha}\right) + \varepsilon_l + O\left(\frac{\left|\mathcal{F}_t^b\right|_C}{n}\right) + \varepsilon_{tb}
\end{aligned}
\tag{16}
$$

**Combining Eqn. (15) and Eqn. (16), it is necessary to satisfy Eqn. (17); otherwise, cross-modal KD would not outperform Unimodal KD.**

$$
\underbrace{O\left(\frac{\left|\mathcal{F}_t^a\right|_C}{n}\right)}_{estimation} + \underbrace{\varepsilon_m + \varepsilon_{ta}}_{approximation} \leq \underbrace{O\left(\frac{\left|\mathcal{F}_t^b\right|_C}{n}\right)}_{estimation} + \underbrace{\varepsilon_l + \varepsilon_{tb}}_{approximation}
\tag{17}
$$

Observing Eqn. (17), we can find that it consists of two components: estimation and approximation error. Regarding the estimation error, it is mainly determined by model capacity and data modality strength. According to the assumption conditions in Sec. 3.1, when we disregard the differences in model capacity and data strength, the estimation error between cross-modal and unimodal KD is not significantly different. Therefore, we will now focus on discussing the approximation error.

Further analysis reveals that concerning the approximation error term, it reflects the disparity between model outputs and the true target. In KD, this difference is primarily determined by the loss function (Eqn. (4). When we use CE and KL divergence as loss functions for model parameter updates, the approximation error introduced by CE is not significantly different for cross-modal and unimodal KD since both use the same labels for teachers and students. However, data distribution shifts lead to inconsistencies in the model output solution space, resulting in varying approximation errors due to KL divergence. Therefore, we will now focus on the approximation error caused by KL divergence. Therefore, we will now focus on the approximation error induced by KL.

In the case of Unimodal KD, the KL divergence is:

$$
KL_{uni}(\mathcal{F}_t||\mathcal{F}_s) = -\sum_{x_i^b} \sigma(f_t(x_i^b)) \log \frac{\sigma(f_s(x_i^b))}{\sigma(f_t(x_i^b))}
\tag{18}
$$

Similarly, in the case of cross-modal KD, the KL divergence is:

$$
KL_{multi}(\mathcal{F}_t||\mathcal{F}_s) = -\sum_{x_i^b} \sigma(f_t(x_i^a)) \log \frac{\sigma(f_s(x_i^b))}{\sigma(f_t(x_i^a))}
\tag{19}
$$

By combining Eqn. (18) and Eqn. (19) and based on the assumption conditions, we can conclude that due to modality differences, the approximation error in cross-modal KD is greater than in Unimodal KD.

$$\begin{aligned}
\varepsilon_m - \varepsilon_l &= KL_{multi}(\mathcal{F}_t||\mathcal{F}_s) - KL_{uni}(\mathcal{F}_t||\mathcal{F}_s) \\
&= KL_{multi}(\mathcal{F}_t||\mathcal{F}_s) - 0 \\
&= KL_{multi}(\mathcal{F}_t||\mathcal{F}_s) \ge 0
\end{aligned} \tag{20}$$

Unimodal KD can be seen as a special case of cross-modal KD (i.e., cross-modal KD without modality shift). **According to Eqn. (20), we can observe that, when the assumption conditions are met, Unimodal KD serves as an upper bound for cross-modal KD.** Thus, we propose the *Solution Space Divergence Hypothesis* to elucidate the impact of modality differences on KD during cross-modal KD.

## B  EXPERIMENTAL SETUP AND MORE RESULTS

### B.1  SYNTHETIC GAUSSIAN IN SEC. 4.2

Assuming two vectors, $x^a \in \mathbb{R}^d$ and $x^b \in \mathbb{R}^d$, constitute a multimodal data pair $(x^a, x^b)$. The feature vectors of $x^a$ and $x^b$ are composed of deterministic features and noise. Initially, we make $x^a$ and $x^b$ identical, and then we fix $x^b$ while altering $x^a$ features. The modification involves gradually replacing deterministic features in xa with other deterministic features. We set $x^b$ as the input data for the student network, while $x^a$ serves as the input data for the teacher network, creating multimodal data with different distribution shifts, as illustrated in Fig. (5). We reduce the dimensionality of the 6 sets of multimodal features to two dimensions for visualization, as shown in Fig. (6).

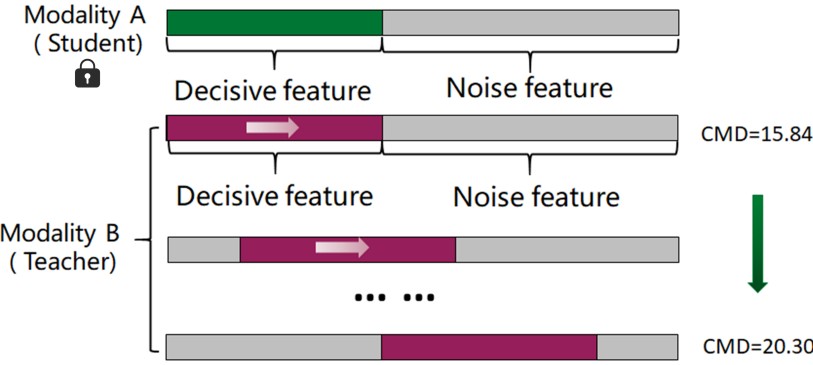

Figure 5: synthetic Gaussian.

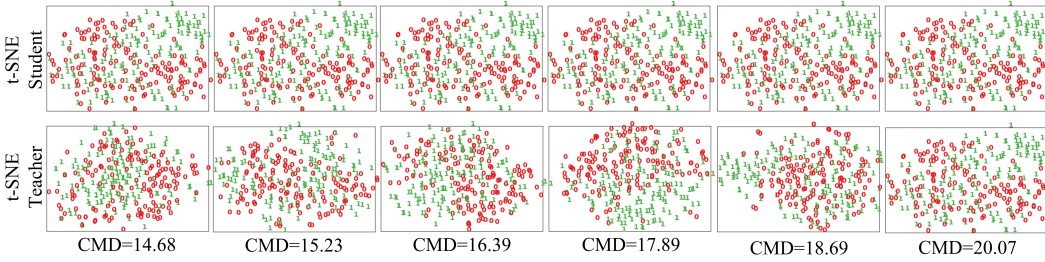

Figure 6: Visualize the 6 sets of multimodal data using t-SNE (Van der Maaten & Hinton, 2008).

### B.2  SKLEARN DATASETS IN SEC. 4.3

**Dataset:** In the process of constructing a multimodal dataset, we adopted two crucial steps. Firstly, by leveraging the 'make_classification' function from the sklearn toolkit, we could effortlessly generate an initial classification dataset, providing a convenient and intuitive starting point for the entire process. Subsequently, by randomly splitting these single-modal features into two, we simulated

features from different modalities. The advantage of this construction method lies in its simplicity and intuitiveness, enabling us to flexibly generate multimodal features and gain a better understanding of the similarities and differences between modalities.

**Further Analysis on the Effectiveness of PSSM:**
This section will further analyze the reasons for the effectiveness of PSSM. Through experiments, we can explain that the effectiveness of the PSSM method lies in its ability to bring about consistency in the solution space of the "Target" section, while preserving the distinctiveness of the solution space for the "None-target" section. The specific analysis is as follows:

We assessed the CMD distance between "Target" and "Non-target" segments of the student network and the evolving probability distribution of the teacher network over network iterations. With the introduction of PSSM in the "Target" segment, probability distributions between teacher and student networks converged, aligning their solution spaces (Fig.(7)(a)). Conversely, in the "Non-target" segment, PSSM increased the separation of probability distributions, preserving distinct solution spaces (Fig.(7)(b)). Table. 7 displays average CMD distances, confirming these trends.

Additionally, we visualized the probability distributions corresponding to the "Target" and "Non-target" segments of both the student and teacher network outputs during the stable training phase.For the "Target" segment, characterized by a binary (0-1) probability distribution, the visualization manifests as a singular point on a plane, as depicted in Fig.(8)(a). Notably, the integration of PSSM results in the convergence of the student's predicted probability distribution toward that of the teacher. This convergence is accentuated with the augmentation of data shift. A similar visualization approach was applied to the probability distributions of the "Non-target" segment. Given its discrete multinomial probability distribution, a spline curve was employed to connect the discrete distribution for enhanced observation, as portrayed in Fig.(8)(b). The graph demonstrates that the introduction of PSSM leads to a divergence between the student's predicted probability distribution and that of the teacher, with this phenomenon becoming more prominent as data shift increases. These observations collectively suggest that the efficacy of the PSSM method lies in fostering consistency in the solution space for the "Target" segment while concurrently preserving distinctiveness in the solution space for the "Non-target" segment.

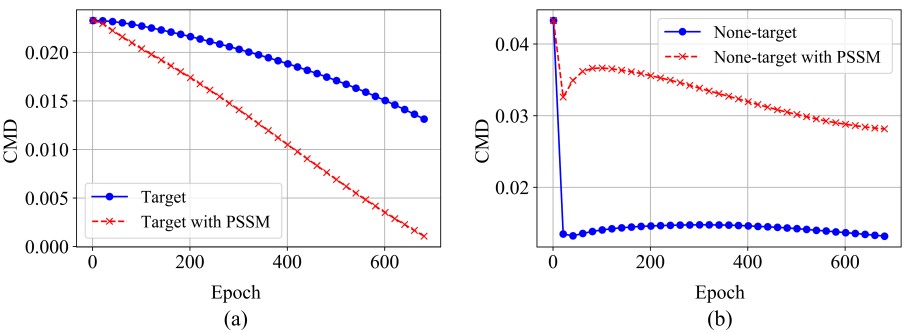

Figure 7: The CMD distance between the output probability distributions of the teacher and student networks. According to (Zhao et al., 2022), we decompose the classification probabilities of the student's teacher network into predictions relevant and irrelevant to the target class. We define the following notations: $\mathbf{b} = [p_t, p_{\backslash t}] \in \mathbb{R}^{1 \times 2}$ represents the binary probabilities of the target class ($p_t$) and all the other non-target classes ($p_{\backslash t}$). Meanwhile, we declare $\hat{\mathbf{P}} = [\hat{p_1}, ..., \hat{p_{t-1}}, \hat{p_{t+1}}, ..., \hat{p_C}] \in \mathbb{R}^{1 \times (C-1)}$ to independently model probabilities among nontarget classes (i.e., without considering the t-th class). The specific calculation method can be found in the (Zhao et al., 2022).

### B.3 MNIST/MNISTM DATASETS IN SEC. 4.4

**Dataset**: MNIST (Modified National Institute of Standards and Technology) is a widely used dataset for handwritten digit recognition. This dataset comprises grayscale images of handwritten digits

Table 7: The average CMD distance between the output probability distributions of the teacher and student networks

| Input CMD / Output CMD×100 | | 4.20 | 4.68 | 5.15 | 5.60 | 5.96 |
|---|---|---|---|---|---|---|
| Target | w/o PSSM | 0.40 | 0.44 | 0.57 | 0.79 | 1.54 |
| | w/ PSSM | 0.44 | 0.84 | 1.34 | 1.44 | 2.23 |
| Non-target | w/o PSSM | 0.34 | 0.42 | 0.40 | 0.64 | 1.23 |
| | w/ PSSM | 1.08 | 2.30 | 2.80 | 3.03 | 4.29 |

ranging from 0 to 9, each corresponding to a label. Each image has a size of 28x28 pixels, with a total of 60,000 training images and 10,000 test images. The MNIST-M (Ganin & Lempitsky, 2015) is created by blending MNIST digits with randomly colored blocks from the BSDS500. To simulate differences from various data sources, we introduced varying levels of noise into the MNIST, as visualized in Fig. (7).

**Implementation Details**: The network consists of a 2-layer MLP. The input data for the student network is MNIST-M, while the input data for the teacher network is MNIST data with added Gaussian noise. This setup allows us to assess the impact of data distribution shift on KD. We conducted experiments with five different noise levels, ranging from 0 to 5. Here, 0 represents data without added noise, corresponding to the original MNIST data, and 5 represents the highest noise level.

### B.4 SEMANTICKITTI DATASET IN SEC. 4.4

**Dataset**: The SemanticKITTI (Behley et al., 2019) is an extension of the KITTI dataset and includes both point cloud data and corresponding images. The point cloud from each frame can be projected onto the image using camera intrinsic and extrinsic matrices, creating a multimodal dataset. For more details on this process, please refer to (Zhuang et al., 2021).

**Implementation Details**: In order to construct two different data distribution disparities, we adopted the following approaches: (1) Images as input data for the teacher network and point clouds as input data for the student network, as depicted in Fig. (8 (a)); (2) Fusion features of images and point clouds as input data for the teacher network, with point clouds as input data for the student network, as shown in Fig. (8 (b)). We consider that the latter has a smaller data distribution shift compared to the former since it incorporates the input data of the student network.

## C ADDITIONAL COMPARISONS

In order to further illustrate the superiority of the PSSM method proposed in Sec.3.3, we compared it with some classical single-modal knowledge distillation methods.

The competing methods included are:

- *Features:* FitNet (Romero et al., 2014), Contrastive Representation Distillation (CRD) (Tian et al., 2019), Relational Knowledge Distillation (RKD) (Park et al., 2019), Probabilistic Knowledge Transfer for deep representation learning(PKT) (Passalis & Tefas, 2018) , Similarity-Preserving KD (SP) (Tung & Mori, 2019).
- *logits:* Knowledge Distillation (KD) (Hinton et al., 2015), decoupled Knowledge Distillation (DKD) (Zhao et al., 2022).

**Comparison results on the Gauss dataset:** Drawing upon empirical findings, our approach demonstrates superior performance across various scenarios, with the only exception being a marginal decrease in effectiveness compared to the SP method, particularly evident when encountering minimal disparities in data distribution, as illustrated in the Table. 8. Notably, our method surpasses other comparative techniques in all remaining situations. As for methodologies such as Contrastive Representation Distillation (CRD), reliant on contrastive learning, their efficacy is compromised on the

Gauss dataset due to its binary categorization and the scarcity of abundant negative sample pairs. Consequently, a comparative analysis of CRD on the Gauss dataset is omitted from the presented table.

Table 8: Comparison with other knowledge distillation methods on the Gauss dataset.

| Distillation manner | CMD | 15.84 | 16.79 | 17.66 | 18.62 | 19.43 | 20.30 |
|---|---|---|---|---|---|---|---|
| | FitNet | 73.48 | 72.47 | 71.33 | 70.24 | 69.32 | 68.33 |
| | CRD | - | - | - | - | - | - |
| Features | RKD | 73.07 | 70.88 | 68.13 | 64.53 | 61.66 | 58.18 |
| | PKT | 73.51 | 72.58 | 71.38 | 70.14 | 68.95 | 68.42 |
| | SP | **73.53** | 72.57 | 71.37 | 70.13 | 68.92 | 68.41 |
| | KD | 73.51 | 72.57 | 71.38 | 70.15 | 68.95 | 68.44 |
| logits | KL+PSSM | 73.29 | **72.58** | 71.94 | 71.44 | 71.31 | 71.06 |
| | DKD | 73.22 | 72.70 | 71.91 | 71.24 | 70.71 | 70.27 |
| | DKD+PSSM | 72.96 | 72.57 | **72.11** | **71.87** | **71.73** | **71.59** |

**Comparison results on the Sklearn dataset:** Table. 9 displays the results for the Sklearn dataset, indicating that our methodology outperforms all classical approaches Notably, its exceptional performance shines in scenarios marked by pronounced disparities in data distribution. This observation underscores the effectiveness of our approach in mitigating the diminishing impact on distillation performance arising from substantial biases in data distribution.

Table 9: Comparison with other knowledge distillation methods on the Sklearn dataset.

| Distillation manner | CMD | 4.20 | 4.68 | 5.15 | 5.60 | 5.96 |
|---|---|---|---|---|---|---|
| | FitNet | 42.28 | 42.20 | 41.34 | 41.16 | 40.60 |
| | CRD | 30.18 | 29.48 | 29.85 | 29.44 | 29.58 |
| Features | RKD | 40.30 | 39.13 | 35.11 | 32.55 | 33.01 |
| | PKT | 42.38 | 42.39 | 41.95 | 41.65 | 41.39 |
| | SP | 42.39 | 42.39 | 41.96 | 41.64 | 41.39 |
| | KL | 42.61 | 42.28 | 41.83 | 41.68 | 41.53 |
| logits | KL+PSSM | 42.16 | 42.25 | 42.07 | 41.92 | 41.73 |
| | DKD | 46.46 | 45.87 | 42.07 | 41.12 | 40.98 |
| | DKD+PSSM | **48.54** | **48.61** | **46.39** | **44.93** | **44.82** |

**Comparison results on the MNIST/MNISTM:** Based on the experimental results, our method exhibits a slight lag behind single-modal knowledge distillation methods only in cases of minimal data distribution differences. Nevertheless, in all other scenarios, it surpasses the comparative methods, as evidenced in Table. 10. This indicates that the performance enhancement of our PSSM method is not solely attributed to the elimination of erroneously assigned pseudo-labels but rather to the increased likelihood that the inconsistency in the solution spaces between the teacher and student occurs precisely where the teacher's pseudo-labels are incorrect, especially in cases of significant data distribution shift.

Table 10: Comparison with other knowledge distillation methods on the MNIST/MNISTM

| Distillation manner | Noise level | 0 | 1 | 2 | 3 | 4 | 5 |
|---|---|---|---|---|---|---|---|
| | FitNet | **82.78** | 81.97 | 81.00 | 76.32 | 73.70 | 74.13 |
| | CRD | 81.05 | 80.86 | 78.67 | 75.47 | 71.58 | 66.02 |
| Features | RKD | 73.07 | 75.17 | 63.00 | 67.83 | 61.94 | 56.70 |
| | PKT | 82.04 | **81.62** | 80.10 | 77.34 | 74.13 | 73.00 |
| | SP | 82.18 | 81.71 | 80.40 | 77.38 | 75.88 | 74.00 |
| | KL | 82.02 | 81.34 | 79.75 | 76.08 | 72.00 | 71.74 |
| logits | KL+PSSM | 81.71 | 81.60 | **80.63** | 77.69 | **77.20** | **76.84** |
| | DKD | 82.03 | 79.87 | 76.52 | 75.14 | 72.19 | 68.99 |
| | DKD+PSSM | 81.83 | 81.21 | 78.77 | **79.92** | 76.94 | 76.20 |

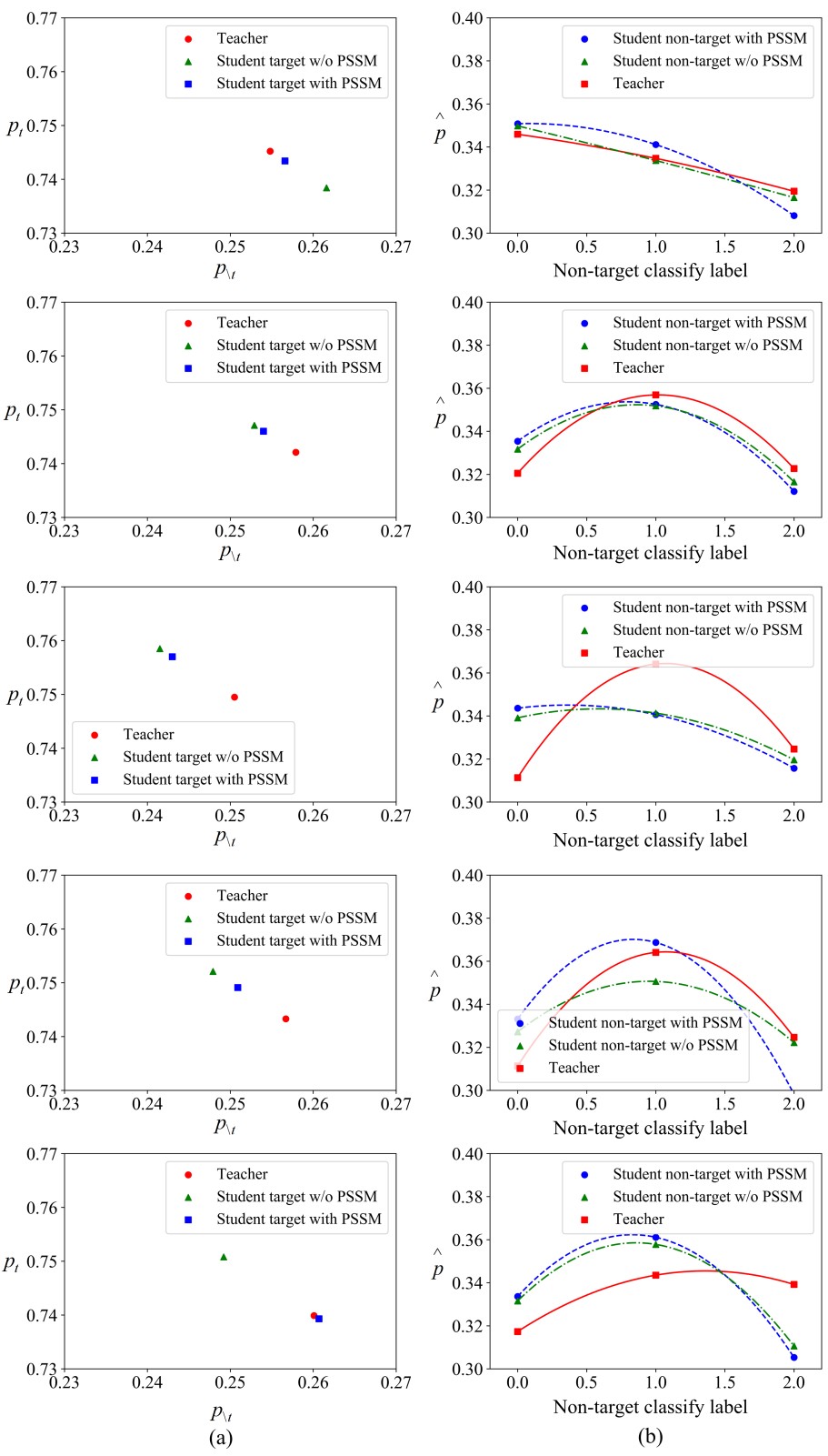

Figure 8: Teacher-student output probability distributions under different data distribution shifts (increasing data distribution shift from top to bottom).

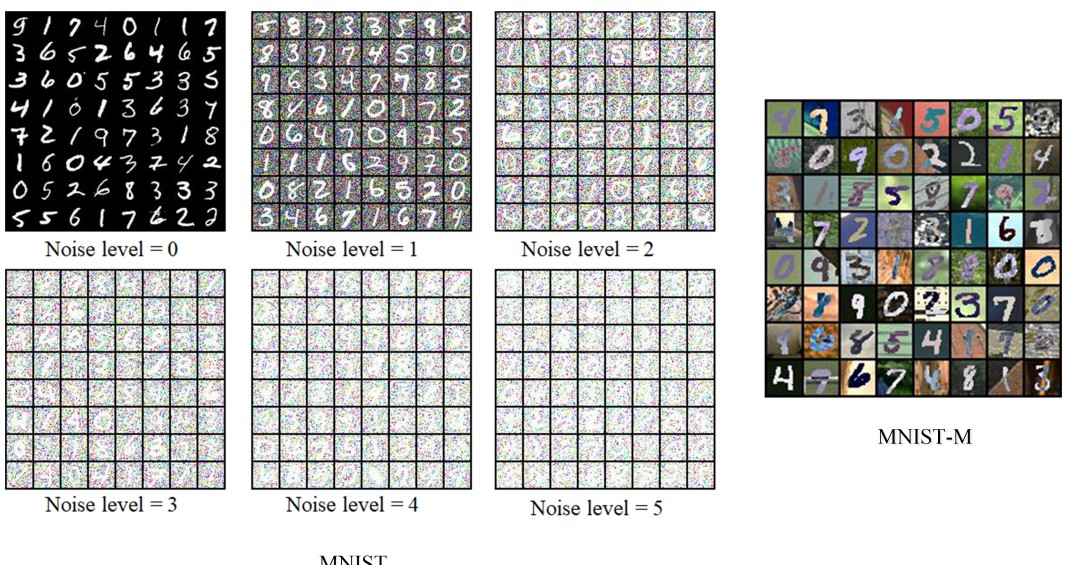

Figure 9: Visualizing the MNIST/MNIST-M dataset.

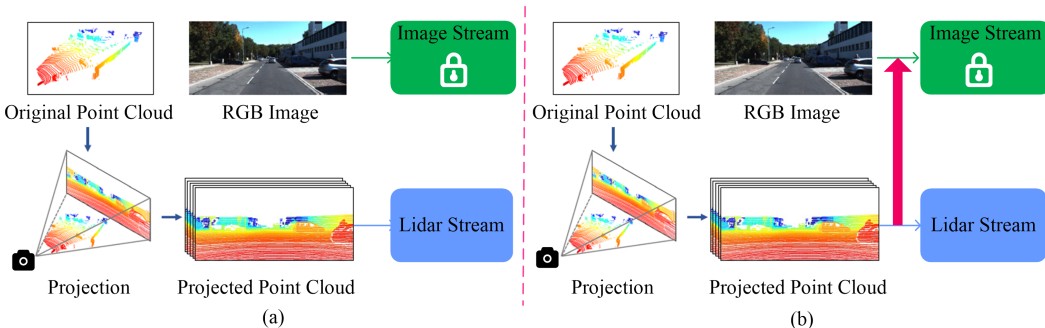

Figure 10: Experiments on the SemanticKITTI dataset. (a) Student: Point Cloud, Teacher: RGB image, (b) Student: Point Cloud, Teacher: Fusion features.

