# OpenReview forum: "Investigating the Impact of Data Distribution Shifts on Cross-Modal Knowledge Distillation"
_ICLR.cc/2024/Conference — Submitted to ICLR 2024_

### Official Review · Reviewer_Pm6d · 2023-10-24

**Soundness:** 3 good
**Presentation:** 2 fair
**Contribution:** 2 fair
**Rating:** 5
**Confidence:** 4

**Summary:**

The paper conducted a comprehensive exploration of cross-modal knowledge distillation and its broader application in multimodal learning. SPDH is introduced to highlight the role if data distribution disparities across modalities in KD effectiveness and PSSM is proposed to mitigate the impact of data distribution shifts on cross-modal KD. Experimental results on four multimodal datasets validate the assumptions and provide directions for future enhancements in cross-modal knowledge transfer.

**Strengths:**

1.	The paper is well-motivated with a focus on the effectiveness of cross-modal KD and its dependence on the distribution shifts in multimodal data.
2.	The theoretical derivations are relatively sufficient and comprehensive.
3.	The experiments validate the method for enhancing the effectiveness of cross-modal KD.

**Weaknesses:**

1.	The paper organization should be optimized, for instance, some detailed derivation can be put in the appendix while it is suggested to use more space for experiments and analysis (Sections 4.3 and 4.4).

2.	The experiments show that the proposed enhanced KD method demonstrates performance improvement compared to the Baseline. However, can it achieve state-of-the-art performance compared to existing cross-modal KD methods (e.g., [a,b]) tailored to specific multimodal tasks?

3.	More qualitative cases are suggested to be provided to better illustrate the effects the proposed enhanced cross-modal KD method has.

[a] Hong Y, Dai H, Ding Y, “Cross-modality knowledge distillation network for monocular 3d object detection”, in European Conference on Computer Vision (ECCV), pp. 87-104, 2022
[b] Wu Z, Li Y, Huang Y, et al, “3D Segmenter: 3D Transformer based Semantic Segmentation via 2D Panoramic Distillation”, in The Eleventh International Conference on Learning Representations (ICLR), 2022.

**Questions:**

The weaknesses mentioned above should be carefully responded in the rebuttal phase.
Besides, there are two more questions:
1.    The figures are not clear enough, especially when zoomed in.
2.    Some typo errors should be carefully checked and modified, like “modaalities” in page 2.

---

> ### Author Response · Authors · 2023-11-23
> **Reply to Reviewer Pm6d**
>
> **Response to Weakness 1:**
> Thank you for your careful review of our paper and valuable suggestions. We highly appreciate your feedback and have made adjustments to the paper structure based on your recommendations.
>
> Under your guidance, we have optimized the paper to ensure a better organizational structure and emphasis on content. Specifically, we have moved some detailed derivations to the appendix, freeing up more space in the main text for experiments and analysis (see the revised version). This adjustment aims to enhance the readability and focus of the paper, ensuring that readers can more directly access experimental results and relevant analyses.
>
> **Response to Weakness 2:**
> Thank you for your review and valuable feedback. We appreciate your concerns and would like to further clarify the intent and contributions of our research.
>
> Firstly, we want to emphasize that the primary goal of our study is to explore the impact of data distribution shifts on cross-modal knowledge distillation, Specifically, the introduction of perceptual divergence distillation loss aims to eliminate inconsistent predictions in the solution space through masking, thereby validating the content of our proposed hypothesis. We observed a substantial increase in our PSSM distillation loss with larger differences in input data distribution, indicating that the inconsistency in the solution space caused by data distribution differences can lead to the failure of cross-modal knowledge distillation. We have expanded the discussion on this aspect in therevised version , and specific details can be found in *Appendix B.2* in the updated paper.
>
> Secondly, we acknowledge that the practical application contributions of our method are relatively limited, and our focus leans more towards theoretical research. We will consider future directions for our work to comprehensively address practical applications and strive to provide more practical solutions for real-world scenarios.
>
> In summary, our work is oriented towards validating specific hypotheses and making theoretical contributions. Once again, thank you for your suggestions and the time you dedicated to the review. We will make every effort to improve and meet your expectations.
>
> **Response to Weakness 3:**
> Thank you for your valuable feedback. To better illustrate the effectiveness of our proposed enhanced cross-modal knowledge distillation method, we have made the following additions:
>
> On the sklearn dataset, we conducted additional analysis to demonstrate the effectiveness of our approach. Specifically, through experiments, we can elucidate that the efficacy of the PSSM method lies in its capability to induce consistency in the solution space of the "Target" section, while preserving the distinctiveness of the solution space for the "Non-target" section. For specific details, please refer to Appendix B.2 in the updated paper.
> We trust that these supplemental experiments provide a more comprehensive understanding of the effectiveness of our approach. Your insightful suggestions have significantly contributed to the refinement of our work.
>
> **Response to Question 1:**
> Thank you for your valuable feedback. We have revised all the figures by using the vector graphics to make it clear when zoomed in. We apologize for any inconvenience caused by the initial lack of clarity in the pictures. We look forward to your improved experience with the updated image quality.
>
> **Response to Question 2:**
> Thank you for your feedback. We have carefully reviewed and corrected spelling errors in the paper, such as "modaalities" . We apologize for any inconvenience caused by these errors. We hope these corrections enhance the quality of the paper.

---

### Official Review · Reviewer_voCm · 2023-10-31

**Soundness:** 3 good
**Presentation:** 3 good
**Contribution:** 3 good
**Rating:** 5
**Confidence:** 3

**Summary:**

This paper examines the influence of data distribution shifts on cross-modal knowledge distillation (KD) and establishes the circumstances in which cross-modal KD surpasses unimodal scenarios. It introduces the Solution Space Divergence Hypothesis (SSDH) to elucidate the difficulties encountered in cross-modal KD and proposes a technique known as the Perceptual Solution Space Mask (PSSM) to tackle substantial disparities in data distribution.

**Strengths:**

The paper includes theoretical analysis and method improvement, which is very good. The SSDH provides an insightful theoretical analysis of how data distribution shifts can lead to divergence between teacher and student solution spaces, hampering cross-modal KD. PSSM is an innovative practical method to enhance cross-modal KD by focusing on output features with smaller solution space differences.  Comprehensive literature review of cross-modal KD and related techniques like data distribution shifts.

**Weaknesses:**

1. Although this article provides a SOLUTION SPACE DIVERGENCE HYPOTHERSIS, the proposed PERCEPTUAL SOLUTION SPACE MASK (PSSM) is  simple and trivial  and plays a similar role to other common knowledge distillation methods.
2. There are few experiments in this paper. Please compare it with more classic single-modal and cross-modal knowledge distillation methods.

**Questions:**

Please refer to the weaknesses.
Please provide evidence of the differences in effectiveness between the PSSM in this article and other classical knowledge distillation methods. Please compare it with more classical unimodal and crossmodal knowledge distillation methods as part of your experiments.

---

> ### Author Response · Authors · 2023-11-23
> **Reply to Reviewer voCm**
>
> **Response to Weakness 1:**
> I extend my sincere appreciation for your invaluable feedback.
>
> Concerning our proposed Perceptual Solution Space Mask (PSSM), it is imperative to emphasize its principal objective, which involves the introduction of a perceptual divergence distillation loss to rectify inconsistent predictions within the solution space through the application of masking. It is essential to clarify that our objective is not primarily focused on presenting a highly innovative knowledge distillation method; instead, it is directed towards substantiating our hypothesis concerning solution space divergence.
>
> As elucidated in the revised paper (Appendix B.2 in the updated paper.), we observe a significant increase in the PSSM distillation loss when confronted with more substantial disparities in input data distribution. Conversely, the degree of enhancement is marginal or even diminishing when faced with minor discrepancies in data distribution. This observation signifies that the incongruity in the solution space, stemming from differences in data distribution, is a pivotal factor contributing to the inefficacy of crossmodal knowledge distillation.
>
> Further elaboration on our research contributions and motivations is provided in Section 1 of the paper, underscoring the central theme of our investigation encapsulated in the paper's title: "Investigating the Impact of Data Distribution Shifts on Crossmodal Knowledge Distillation."
>
> In conclusion, I wish to reiterate my gratitude for your constructive feedback. Your insightful suggestions have significantly enriched the overall quality of the paper.
>
> **Response to Weakness 2:**
> I appreciate your valuable feedback.
>
> We have addressed and incorporated your suggestions, including conducting comparative experiments with five classic single-modal knowledge distillation methods.
> Results demonstrate that, as data distribution differences increase, none of the single-modal methods exhibit effective performance, affirming the universal impact of data shift on cross-modal knowledge distillation.
>
> Specific comparative results are presented in **Tables 8, 9, and 10** for your reference, offering a comprehensive view of our method's performance amidst data distribution differences and emphasizing the persistent challenge in traditional single-modal methods. We have added these experiments on **Page 17-18** in the revised version.
>
> Thank you once again for your insightful feedback, contributing significantly to the paper's improvement.

---

### Official Review · Reviewer_GxY4 · 2023-10-31

**Soundness:** 3 good
**Presentation:** 3 good
**Contribution:** 3 good
**Rating:** 6
**Confidence:** 3

**Summary:**

This paper investigates a weighted distillation loss by cosine similarity between teacher and student networks' logits under cross-modal setting.

**Strengths:**

This paper is well-written with clear assumptions and hypothesis, where later the hypothesis is experimentally validated by using synthetic Gaussian data.

**Weaknesses:**

My biggest confusion is that it is relatively difficult for me to connect data distribution shifts with KL divergence between two different probability distributions. From my understanding, the connection between data distribution shifts (here in this paper, modality differences) and ''solution space'' is a little bit absurd. Looking forward to further clarifications.

And also please see questions.

**Questions:**

1. Please check Eqn. (11), it might be wrong after the = symbol;
2. Why choose the cosine similarity function? Can other functions be used? What are the results?
3. Just curious, for A.3, what are the results if input noisy MNIST to student and use MNIST-M for teacher?

---

> ### Author Response · Authors · 2023-11-23
> **Reply to Reviewer GxY4**
>
> **Response to Weakness :**
> We sincerely apologize for any confusion caused by inaccuracies in our expression. Typically, neural network parameters represent the solution space. However, in our paper, we establish a connection between changes in data distribution (specifically, modal differences) and the "solution space" based on the provided assumptions 1 and 2. Specifically, under the conditions of assumptions 1 and 2, the only distinction between cross-modal knowledge distillation and single-modal knowledge distillation is the difference in data distribution. Thus, we can replace the solution space, originally represented by network parameters, with the output probability distribution after inputting the data into the network. In summary, this connection is often invalid without the prerequisites of the first two assumptions.
>
> Finally, we have revised and refined the expressions related to this section in the paper, making it more rigorous. For detailed information, please **refer to the updatedversion, specifically page 4**. We appreciate once again the constructive feedback you provided, which played a crucial role in enhancing the quality of the paper.
>
>
> **Response to Question 1:**
> Thank you for your meticulous review.We have revised it in the revised version.
>
> **Response to Question 2:**
> Thank you for your attention to the details of our paper.
>
> We chose the cosine similarity function for the following reasons:
>
> (1). **Necessity of Relative Distance:** In comparison to other distance metrics such as JS distance and Euclidean distance, which typically have an infinite range, our PSSM method requires setting thresholds based on distribution distances within a finite range. Therefore, we opted for cosine similarity rather than absolute distance to characterize distribution differences.
>
> (2). **Advantages of Cosine Similarity:** The range of cosine similarity is easily scaled between 0 and 1, allowing us to quantify distribution differences on a relative and standardized scale. Specifically, when cosine similarity is 0, it indicates no distribution difference, and when it is 1, it signifies maximum distribution difference. This enables us to comprehend and measure cross-modal data distribution differences more clearly.
>
> Considering these two reasons, we chose the cosine similarity function.
>
> **Response to Question 3**
> The reason we did not use "input noisy MNIST to student and use MNIST-M for teacher" is that, with the introduction of noise, the teacher's accuracy would sharply decline, while the student network's accuracy would remain high. In this scenario, the student would not effectively acquire knowledge from the teacher. To address your inquiry, we conducted additional experiments by employing noisy MNIST as input for the student network and MNIST-M as input for the teacher network. The results are shown in the table below:
>
> | Noise Level | Teacher Accuracy | Student w/o KD | Student KD DKD | Student KD DKD with PSSM |
> |:------------:|:----------------:|:---------------:|:---------------:|:------------------------:|
> | 0            | 74.87           | 92.98           | 88.46 (-4.52 ± 0.18) | **92.15** (-0.83 ± 0.38)      |
> | 1            | 37.77           | 92.98           | 70.28 (-22.70 ± 7.20) | **90.02** (-2.96 ± 2.00)  |
> | 2            | 19.77           | 92.98            | 70.87 (-22.14 ± 7.31) | **79.28** (-13.70 ± 0.80) |
> | 3            | 15.11            | 92.98            |67.52 (-25.96 ± 5.46) | **77.67** (-15.31 ± 4.03) |
> | 4            | 12.42            | 92.98            |71.36 (-21.62 ± 0.80) | **74.86** (-18.12 ± 4.29) |
> | 5            | 11.36            | 92.98            |64.36 (-28.62 ± 2.90) | **72.86** (-20.12 ± 3.49) |

---

### Official Review · Reviewer_NCD3 · 2023-11-10

**Soundness:** 3 good
**Presentation:** 3 good
**Contribution:** 3 good
**Rating:** 6
**Confidence:** 3

**Summary:**

This work explored cross-modal KD (knowledge distillation) in multimodal learning. First, the hypothesis of solution space divergence (SPDH) is introduced to show that the success in cross-modal KD is decided by the data distribution shift. Then an effective method called PSSM (perceptual solution space mask) is proposed to enhance cross-modal KD. Experimental results on four popular datasets verify the effectiveness of the proposed hypothesis and method.

**Strengths:**

**Originality**: The paper proposes a hypothesis, SSDH, to show the key factor in cross-modal KD and a method, PSSM, to tackle the degradation in KD due to multimodal data. Both SSDH and PSSM are instructive.

**Quality**: The paper provides extensive experimental evaluations of the proposed hypothesis and method.

**Clarity**: The paper also provides sufficient background information and related work to situate the contribution of the proposed hypothesis and method in the context of existing literature on cross-modal KD, KD analysis, and distribution shifts.

**Significance**: The paper has established the conditions under which cross-modal KD outperforms unimodal scenarios. This is very important for future research.

**Weaknesses:**

**Symbol List in the Appendix**:
It would be beneficial to include a comprehensive symbol list in the Appendix. This addition will enhance the clarity of the notation used throughout the paper.

**Table Placement**:
Consider moving some experimental results from the Appendix, specifically Tables 2, 5, and 6, into the main paper. On the other hand, certain equations from Sec. 3.2 might be more appropriately placed in the Appendix to streamline the main content.

**Formatting Improvements**:
The format of the paper requires further attention. Specifically, the use of `\cite` and `\citet` appears confusing. A consistent and clear citation style should be maintained throughout the manuscript.

**Questions:**

Please address the weaknesses above.

---

> ### Author Response · Authors · 2023-11-23
> **Reply to Reviewer NCD3**
>
> **Response to Weakness 1:**
> Thank you for providing constructive and meaningful suggestions. We will include a comprehensive symbol list in the appendix to enhance the clarity of symbol usage throughout the paper. This addition will offer readers a more comprehensive reference, ensuring they can accurately understand the symbols and their meanings used in the paper. The specific details can be found in the updated PDF paper in the *appendix A.*
>
> | Symbol                  | Definition                                       |
> |-------------------------|--------------------------------------------------|
> | ${x}^{a}$               | Data from modality 'A'                           |
> | ${x}^{b}$               | Data from modality 'B'                           |
> | ${\mathcal{F}_{t}}$     | Teacher function (trained on ${x}^{a}$)          |
> | ${\mathcal{F}_{s}}$     | Student function (trained on ${x}^{a}$ or ${x}^{b}$)  |
> | $T$                     | Temperature parameter                            |
> | ${y}_{i}$               | The true hard labels                             |
> | ${s}_{i}$               | Soft predictions                                |
> | ${\varepsilon}$         | Approximation error                              |
> | $O(\cdot)$              | Estimation error                                 |
> | $\sigma$                | The softmax operation                            |
> | ${{\left| \cdot \right|}_{C}}$ | Some function class capacity measure   |
> | $n$                     | The number of data points                        |
>
> **Response to Weakness 2:**
> Thank you for your meticulous review of our manuscript and your invaluable suggestions. We greatly appreciate the insightful feedback provided, and, in response, we have implemented adjustments to the paper's structure in accordance with your recommendations.
>
> Under your guidance, we have refined the paper to achieve an improved organizational structure and enhanced emphasis on content. Notably, detailed derivations have been relocated to the appendix, thereby liberating additional space in the main text dedicated to experiments and analyses (refer to the Rebuttal Revision PDF). This strategic adjustment is intended to heighten the paper's readability and focus, ensuring that readers can more directly access experimental results and pertinent analyses.
>
> **Response to Weakness 3:**
> Thank you very much for your meticulous review. When the authors or the publication are included in the sentence, the citation should not be in parenthesis using \verb|\citet{}|. Otherwise, the citation should be in parenthesis using \verb|\citep{}|. We have thoroughly checked and corrected all citations in the paper to ensure consistency. The revised results can be found in the updated PDF version of the paper. Once again, we appreciate your valuable suggestions, which have significantly enhanced the standardization of the paper.

---

### Meta-Review · Area_Chair_9JXJ · 2023-12-04

**Metareview:**

This paper aims at the problem of cross-modal knowledge distillation and investigates the impact of data distribution shift for the KD performance. Through simulation experiments, the authors reveal the data distribution shifts play a significant role for performance of the student model. To address this issue, the authors proposed a so called "perceptual solution space mask" to enhance the effectiveness of cross-modal KD.

**Justification For Why Not Higher Score:**

- The investigated problem is narrow.
- The experiments are only conducted on small-scale toy datasets.
- Comparison with SOTA KD methods is missing.

**Justification For Why Not Lower Score:**

NA

---

### Decision · Program_Chairs · 2024-01-16

Reject